# Antitumor Effect of Iscador on Breast Cancer Cell Lines with Different Metastatic Potential

**DOI:** 10.3390/ijms24065247

**Published:** 2023-03-09

**Authors:** Bozhil Robev, Ivan Iliev, Iana Tsoneva, Albena Momchilova, Alexandrina Nesheva, Aneliya Kostadinova, Galya Staneva, Biliana Nikolova

**Affiliations:** 1Department of Medical Oncology, University Hospital “Sv. Ivan Rilski”, 15 Acad. Ivan Geshov Blvd., 1431 Sofia, Bulgaria; 2Institute of Experimental Morphology, Pathology and Anthropology with Museum, Bulgarian Academy of Sciences, Acad. G. Bonchev Str., Bl. 25, 1113 Sofia, Bulgaria; 3Institute of Biophysics and Biomedical Engineering, Bulgarian Academy of Sciences, Acad. G. Bonchev Str., Bl. 21, 1113 Sofia, Bulgaria

**Keywords:** Iscador, antiproliferative activity, cytotoxicity, actin cytoskeleton, migration potential, lipid order, zeta potential

## Abstract

Studies were performed for the first time on the effect of Iscador Qu and Iscador M on phototoxicity, cytotoxicity, antiproliferative activity, changes in ξ-potential of cells, membrane lipid order, actin cytoskeleton organization and migration on three breast cancer lines with different metastatic potential: MCF10A (control), MCF-7 (low metastatic) and MDA-MB231 (high metastatic) cells. The tested Iscador Qu and M did not show any phototoxicity. The antiproliferative effect of Iscador species appeared to be dose-dependent and was related to the metastatic potential of the tested cell lines. A higher selectivity index was obtained for Iscador Qu and M towards the low metastatic MCF-7 cell line compared to the high metastatic MDA-MB-231. Iscador Qu demonstrated higher selectivity for both cancer cell lines compared to Iscador M. The malignant cell lines exhibited a decrease in fibril number and thickness regardless of the type of Iscador used. The strongest effect on migration potential was observed for the low metastatic cancer cell line MCF-7 after Iscador treatment. Both Iscador species induced a slight increase in the percentage of cells in early apoptosis for the low and high metastatic cell lines, MCF-7 and MDA-MB-231, unlike control cells. Changes in the zeta potential and membrane lipid order were observed for the low metastatic MCF-7 cell line in contrast to the high metastatic MDA-MB-231 cells. The presented results reveal a higher potential of Iscador as an antitumor agent for the low metastatic cancer cell line MCF-7 compared to the high metastatic one. Iscador Qu appears to be more potent compared to Iscador M, but at this point, the exact mechanism of action is still unclear and needs further investigations.

## 1. Introduction

Early diagnosis and treatment of breast tumors have undergone rapid progress in the recent years, and yet they remain the most frequent cancer among women and a leading cause of mortality [1,2,3]. Breast cancer is heterogeneous group of genetically predisposed diseases with diverse clinical futures [4,5]. Classification of breast cancer could be done into five subtypes: luminal A, luminal B, HER2, basal and normal [6,7]. Each subtype has a different prognosis and response to antitumor treatment. 

Significant research effort is focused on the causes of breast cancer onset and the mechanisms of progression aiming at development of new clinical protocols with lower overall toxicity to healthy tissues. The main conventional method of treating breast cancer remains chemotherapy, despite the low selectivity of the drugs used, thus affecting and damaging both a high percentage of healthy cells and the target cancer cells.

The demand for drugs with higher efficacy and lower side effects, mainly from natural sources such as plants, has grown exponentially in recent years. Plants produce a wide range of chemical compounds called secondary metabolites in response to stress. Secondary metabolites exhibit therapeutic effects on human health such as anti-inflammatory, anticancer, antibacterial, antiviral, antioxidant among others [8].

Mistletoe (*Viscum album L.*) from the Santalaceae family is an evergreen, perennial, hemiparasitic plant that absorbs water and nutrients from the host tree. The importance of *V. album* is related to its pharmacological effects such as anticarcinogenic [9,10] antidiabetic, antioxidant [11], blood pressure-lowering [12], sedative [13], antibacterial [14], antiviral [15], pro-apoptotic [16], immunomodulatory [17] and cytotoxic [18] proved by many studies [19]. The variety of therapeutic applications results from the rich chemical composition of *Viscum* species, which largely depend on the host species. The mistletoe plants growing on different host trees determine the type of mistletoe extract like oak for Iscador Qu and apple for Iscador M. Lectins, viscotoxins, flavonoids, phenolic acids, sterols, lignans, terpenoids, phenylpropanoids, alkaloids and fatty acids are among the active components in the mistletoe extract [20]. Cytotoxic glycoproteins, the mistletoe lectins, are considered as the most active component of mistletoe extracts. It is suggested that the antitumor properties of mistletoe extracts originates from lectins which are able to stimulate effector cells of the innate and adaptive immune system such as dendritic cells, macrophages, natural killer cells, as well as B and T lymphocytes [21,22]. Moreover, lectins show direct growth inhibition and cell death induction in tumor cells by causing apoptosis or direct necrotic effects [23,24,25,26].

Mistletoe has been used as complementary anticancer therapy for more than 50 years in German-speaking countries [27]. At present the standardized extracts from the white mistletoe plant are the most promising as additional drug therapy and are used among patients with various types of cancer. The complementarity of the therapy can be applied as adjuvant before, during or after chemotherapy. Mistletoe extract treatment against breast cancer has been recommended due to its minimal side effects [28].

The main compounds with anticancer activity isolated from *Viscum* species are lectins [29] and viscotoxins [30]. However, it has been reported that Mistletoe extract is more potent comparing isolated components from the extract [31]. Despite promising results presented in several clinical trials and biological studies [16,32,33], little is known about the exact mechanism of action of mistletoe extracts on cancer cells. As the plasma membrane is the first target of drugs it is important to analyze the interaction of the extract with plasma membranes of different types of cancer cell lines depending on their metastatic potential. Basic physicochemical and biophysical characteristics of the plasma membrane such as lipid order and membrane potential are starting point for elucidating the mechanism of drug-cell interaction, drug-cell penetration and, accordingly, the mechanism of drug cytotoxicity. It is of utmost importance to establish the interaction of the Mistletoe extract with the actin cytoskeleton, which is assumed to be involved in a number of regulatory processes, affecting cell shape, motility, transport, interactions with the environment, and metastatic potential of cancer cells. 

The goal of this study is to assess and compare the antitumor activity of two types of white Mistletoe extract, Iscador Qu and Iscador M, on a model of breast cancer composed of three human lines with different metastatic potential: the non-tumorigenic MCF-10A; and two cancerous cell lines with different metastatic potential MCF-7 and MDA-MB231. 

The specific interactions of Iscador Qu and Iscador M with three cell lines will be characterised by zeta potential and membrane lipid order measurements. Furthermore, the changes in actin cytoskeleton and migration potential will be compared between studied cell lines and both extracts. A correlation between electrical, membrane, cytoskeleton changes and cytotoxicity induced by Mistletoe extracts on the three cell lines will be done to suggest a possible mechanism of action of both drugs separately. These data will reveal which type of Iscador is more potent as a drug against breast cancer cells *in vitro* and will shed more light on the mechanisms of drug resistance in high metastatic MDA-MB231 cell line.

## 2. Results

### 2.1. Cyto- and Photo-Toxicity

Exposure to daylight can convert some organic compounds to phototoxins due to changes in the energy state or molecular degradation. The evaluation of phototoxicity is an indicator of the photostability of the extracts as well as evidence that their effect on cell viability is due to cytotoxicity alone and not phototoxicity.

Iscador M and Qu white mistletoe extract was investigated by phototoxicity test on BALB 3T3/23 clone A31 (mouse embryonic fibroblasts) cell line. The Neutral Red Reduction (NRU) test is based on comparison of the cytotoxicity of Iscador M and Qu in the presence and absence of non-cytotoxic exposure to UVA/vis light. BALB 3T3 cells were treated with Iscador M and Qu separately in a wide concentration range from 0 to 4000 µg/mL. The obtained results are shown in Figure 1.

The results of cytotoxicity and phototoxicity studies showed that the effect of Iscador Qu on the BALB 3T3/23 cell line is dose-dependent. The tested substance does not show phototoxicity, as the cytotoxicity and phototoxicity curves as a function of Iscador Qu concentration did not differ by more than 10%. This makes Iscador substances suitable for subsequent experiments to establish the antiproliferative activity of plant extracts isolated from white mistletoe with trade names Iscador Qu and Iscador M.

### 2.2. Antiproliferative Activity of Iscador Qu and Iscador M

To investigate the antiproliferative activity, two cancer lines differing in their metastatic potential, MCF-7 (low metastatic) and MDA-MB-231 (high metastatic), and a non-tumorigenic epithelial line MCF-10A were tested. Iscador Qu and Iscador M were used in a wide concentration range from 0 to 1000 µg/mL (Figure 2). 

Cell survival of all studied cell lines was measured after 72 h. Antiproliferative activity as a function of drug concentration produced typical sigmoidal curves. The IC_50_ values of Iscador Qu and Iscador M on MCF-7 and MDA-MB-231 were calculated and are presented in Figure 3B. Iscador Qu showed stronger antiproliferative activity compared to Iscador M. Iscador M reached 50% antiproliferative effect at two-fold higher concentrations compared to Iscador Qu, which was observed for the three cell lines studied. The lowest Iscador Qu concentration at which antiproliferative activity was detected was IC_50_ = 40.47 μg/mL, which was found for the MCF-7 cell line.

All IC_50_ values for the cancer cell lines (MCF-7 and MDA-MB-231) were lower compared to the control non-cancer cell line (MCF-10). This implies that Iscador Qu and Iscador M are less cytotoxic to non-tumorigenic cells than to cancer cell lines (Figure 3B).

### 2.3. Selectivity of Iscador Qu and Iscador M to Malignant Cells

To calculate the selectivity index (SI), the IC_50_ data were used. The SI represents the potential of a given substance to be used as an antitumor medication. The following formula was used for SI calculations:$$\mathrm{SI}\;\left(selectivity\;index\right)=\frac{{\mathrm{IC}}_{50}^{noncancer\;cells}}{{\mathrm{IC}}_{50}^{cancer\;cells}}$$

Based on the results presented in Table 1, both Iscador Qu and Iscador M can be defined as substances with selectivity towards cancer cells. From two to three-fold higher SI was obtained for both Iscadors, Qu and M, towards the low metastatic MCF-7 cell line compared to the high metastatic MDA-MB-231 one. Iscador Qu demonstrated higher selectivity compared to Iscador M towards both cancer cell lines. The highest SI (7.69) was obtained for Iscador Qu towards MCF-7 cell line.

### 2.4. Changes in Membrane Lipid Order of the Studied Cell Lines under Iscador Treatment

First step in this part of the study was determination of the membrane lipid order of the non-treated cells. Laurdan spectra of the three cell lines are presented in Figure 4A. Laurdan emission shifts to the longer wavelengths for malignant cell lines compared to non-cancerous MCF-10A, which clearly implies lipid order decrease in the cancer cell membranes. The difference between the control cell line spectrum (MCF-10A) and the tumorigenic ones is well pronounced whereas the difference in spectra between MCF-7 and MDA-MB-231 is less distinct (Figure 4B). Three independent experiments were performed with six measurements per experiment which provides an average spectrum from 18 measurements per cell line. Such design of experiments allows to perform statistical analysis of the obtained GP values, in particular to distinguish the lipid order of the malignant cell lines. The GP values of the three cell lines are presented in Figure 5 showing that they vary from 0.4 to 0.6. These GP values, i.e., membrane lipid orders, are characteristic for liquid-ordered membranes. A statistically significant difference in GP values was found for the three cell lines. The membranes of MCF-10A were most ordered, whereas the tumorigenic MCF-7 and MDA-MB-231 were less ordered. The calculated GP values for malignant cells confirm the literature reports that membranes of malignant cells are more fluid compared to the membrane of non-cancerous cells. The high metastatic cell line MDA-MB-231 showed a higher lipid order compared to the low metastatic MCF-7, possibly due to the higher cholesterol content found in MDA-MB-231 line compared to MCF-7 [34].

To investigate changes in membrane lipid order after Iscador treatment, we selected two different concentrations for each cell line, IC_50_ and a lower one. These were 100 µg/mL and 50 µg/mL for MCF-10A, 40 µg/mL and 20 µg/mL for MCF-7, and 80 µg/mL and 40 µg/mL for MDA-MB-231. Iscador was able to affect the lipid order of the cell membranes of non-cancer cells (Figure 5). Statistically significant differences were observed between treated and non-treated MCF-10A cells at the higher concentration of Iscador Qu (100 µg/mL) and at both tested concentrations of Iscador M (50 µg/mL and 100 µg/mL) (Figure 5A). At these concentrations, Iscador was able to increase the membrane lipid order of MCF-10A cells.

The hormone-dependent cell line MCF-7 responded to the treatment with the higher concentration of Iscador (40 µg/mL) (Figure 5B). The GP values of Iscador Qu treatment were lower relative to the untreated control, indicating that this concentration had a disordering effect on the cell membranes, whereas the opposite effect was observed for Iscador M treatment (Figure 5B), similar to the changes in the MCF-10A cell line where IC_50_ is much higher. In the triple negative cell line MDA-MB-231, after Iscador Qu or Iscador M treatment, no statistically significant differences in the membrane lipid order were observed (Figure 5C) at the two studied concentrations.

### 2.5. Changes in ζ-Potential of the Studied Cell Lines under Iscador Treatment

Iscador Qu or M treatment of tumorigenic and non-tumorigenic cell lines demonstrated that both types of Iscador did not affect the ζ-potential of MCF-10A and MDA-MB-231 cell lines (Figure 6). In contrast, the applied statistical analysis distinguished the changes in ζ-potential of treated from non-treated MCF-7 cells with *p* < 0.05. The Iscador- treated cells exhibited more negative values of the potential, as shown in Figure 6. Furthermore, the effect of Iscador Qu treatment was higher compared to Iscador M (Figure 6) as the ζ-potential values shifted to higher negative ones. The observed changes in ζ-potential values of the treated cell lines with Iscador (Qu and M) were in accordance with the established antiproliferative activity rank of the studied cell lines: MCF-7 > MDA-MB-231 > MCF-10A. The lowest IC_50_ value was found for Iscador Qu-treated MCF-7 cells (IC_50_ = 40.47 ± 3.11), which corresponds to the largest ζ-potential decrease versus Iscador M-treated ones at the same concentration (Figure 6).

### 2.6. Effect of Iscador Qu and Iscador M on Actin Cytoskeleton Organization and Migration

#### Actin Cytoskeleton

The morphological investigations of the studied cell lines were performed after actin staining. The changes in the localization of the actin cytoskeleton of the three cell lines was examined after 24 h treatment with Iscador Qu and Iscador M. Untreated control cells (MCF-10A) showed intact actin filaments with a lot of long-distal stress fibers and polygonal morphology typical for epithelial cells (Figure 7A). After Iscador treatment, the actin cytoskeleton organization was changed; the actin filaments (stress fibers) decreased in number and thinned out (Figure 7B,C). No changes in cell shape were perceived after treatment with Iscador Qu and M. Thus, neither of the studied Iscador Qu and M extracts significantly affected the integrity of the actin cytoskeleton of MCF-10A cells. However, radial symmetry of the actin cytoskeleton was observed for MCF-10A cells after Iscador Qu treatment. Thus, Iscador Qu was able to “switch on” the actin cytoskeleton by changing the actin fibers from long-distal to radial distribution.

The untreated cells of both cell lines, MCF-7 and MDA-MB-231, clearly showed well-formed stress fibers (follow the arrows in Figure 8A and Figure 9A) located along the length of the cell body. A significant change in the cytoskeleton of tumorigenic cell lines, MCF-7 (Figure 8) and MDA-MB-231 (Figure 9), was detected after treatment with IC_50_ of Iscador Qu or Iscador M. The actin filaments were more marked in the focal adhesion junction (Figure 8B and Figure 9B). This finding was more pronounced for Iscador Qu treatment for the MCF-7 and MDA-MB-231 cell lines. Iscador Qu and M extracts exhibited a tendency to disrupt the actin filaments of cancer cells. The absence of filopodia and lamellipodia in the treated cell lines is a sign of decreased motility of the cancer cells. Depolymerization and diffuse distribution of the actin filaments were also observed (Figure 8B–H and Figure 9B–H). In response to Iscador treatment, a regular organization of the actin filaments was no longer detected. Such changes in the organization of the actin cytoskeleton give rise to weak cell adhesions and affect cell survival.

### 2.7. Cell Migration Assay

The next step of our study was to analyze changes in the migration capacity of cells after treatment with Iscador (Figure 10). Cell-to-cell interactions and cell migration were assessed using a wound healing assay. Cell migration is a hallmark of cancer invasion and metastasis, wound repair, angiogenesis, etc. In vitro analysis of cell migration is useful to follow the alterations in cell migratory potential as a response to experimental treatments. In our study, the migration potential was analyzed 24 h after treatment with Iscador Qu and M. The results presented in Figure 10 show minimal differences in cell migration after treatment of the non-tumorigenic cell line MCF10A. About 25% wound replacement of treated cells and 30% of untreated MCF10A was detected (Figure 10A,B). Similar results were obtained after treatment of the high metastatic cell line MDA-MB-231 (Figure 10E,F). A statistically significant difference in the wound area is found for Iscador Qu compared to control for the low metastatic breast cancer cell line MCF-7 (Figure 10C,D). These results are in good agreement with the data obtained from the viability test (Figure 2).

### 2.8. FACS Analysis

Treatment of the tested cell lines with Iscador Qu or Iscador M resulted in a slight increase in the percentage of cells in early apoptosis (Figure 11). The highest increase was reported in the low metastatic cell line MCF-7. An increased percentage of cells in late apoptosis was observed after Iscador M treatment of the low metastatic MCF-7 cell line. 

## 3. Materials and Methods

### 3.1. Drugs

*Viscum album* L. extracts: Iscador Qu 20 mg/mL solution for injection (Iscador AG, Lörrach, Germany), host plant oak, and Iscador M 20 mg/mL injection solution (Iscador AG, Lörrach, Germany), host plant apple. Since Iscador is an approved medicinal product for intravenous administration to patients, it is dissolved in saline. During in vitro experiments, Iscador was diluted in cell culture media. Thus, it was not necessary to control for the solvent effect.

### 3.2. Cell Lines

All cell lines, breast cancer MCF-10A, MCF-7 and MDA-MB-231, and BALB/c 3T3 clone A31 (mouse embryonic fibroblasts) were purchased from the American Type Culture Collection—ATCC (Manassas, VA, USA). The MCF-7, MDA-MB-231 and BALB/c 3T3 clone A31 were cultivated in Dulbecco’s modified Eagle’s medium (DMEM) supplemented with 10% fetal bovine serum (FBS), 1% sodium pyruvate and 1% MEM non-essential amino acids (NEAA) without antibiotics. The non-tumorigenic breast cell line (MCF-10A) was cultivated in DMEM medium (Sigma-Aldrich, St. Louis, MO, USA) supplemented with 10% FBS, 1% sodium pyruvate, 1% non-essential amino acids (NEAA), 20 ng/mL human epithelia growth factor (hEGF), 10 μg/mL insulin and 0.5 μg/mL hydrocortisone without antibiotics. All cell lines were maintained in a humidified atmosphere with 5% CO_2_ at 37 °C.

### 3.3. Cytotoxicity and Phototoxicity Testing

BALB/3T3 cells were cultured in 75 cm^2^ tissue culture flasks. Cytotoxicity/phototoxicity was assessed by validated BALB/3T3 clone A31 Neutral Red Uptake Assay (3T3 NRU test) [35,36]. Cells were plated in a 96-well plate at a density of 1 ×10^4^ cells/100 µL/well, incubated for 24 h and after that washed. The studied compounds were applied in a wide concentration range. For phototoxicity tests, irradiation with 2.4 J/cm^2^ dose was followed by incubation for additional 24 h. The absorption was measured on a TECAN microplate reader (TECAN, Grödig, Austria) at wavelength 540 nm.

Cytotoxicity/phototoxicity were expressed as CC_50_/PC_50_ values (concentrations required for 50% cytotoxicity/phototoxicity), calculated using non-linear regression analysis (GraphPad Software, San Diego, CA, USA). The CC_50_ values can be used to calculate the PIF (photo-irritancy factor) for each test substance, according to the following formula:$$\mathrm{PIF}\;\left(Photo-Irritancy\;Factor\right)=\frac{Cytotoxicity\;\left({\mathrm{CC}}_{50}\right)}{Phototoxicity\;\left({\mathrm{PC}}_{50}\right)}$$

The statistical analysis included application of one-way ANOVA followed by Bonferroni’s post hoc test. *p* < 0.05 was accepted as the lowest level of statistical significance. All results are presented as mean ± SD.

### 3.4. In Vitro Antiproliferative Activity

For this study the standard MTT-dye reduction assay was used to test the antiproliferative activity [37]. The method is based on the metabolism of the tetrazolium salt MTT to insoluble formazan. The formazan absorption at λ = 540 nm was registered using a TECAN microplate reader (TECAN, Grödig, Austria). The absorption measured is an indicator of cell viability and metabolic activity. Antiproliferative activities were expressed as IC_50_ values (concentrations required for 50% inhibition of cell growth), calculated using non-linear regression analysis (GraphPad Software, San Diego, CA, USA).

The statistical analysis included application of one-way ANOVA followed by Bonferroni’s post hoc test. *p* < 0.05 was accepted as the lowest level of statistical significance. All results are presented as mean ± SD. All experiments were performed in triplicate.

### 3.5. Assessment of the Plasma Membrane Lipid Order

To study the plasma membrane lipid order, fluorescent probe 6-Dodecanoyl-2-dimethyl-aminonaphthalene (Laurdan) was purchased from Sigma Aldrich (Saint Louis, MO, USA). The fluorescent probe was solubilized in DMSO to achieve a 5 mM stock concentration. 

MCF-10A, MCF-7 and MDA-MB-231 at an overall density of 1 × 10^6^ cells/mL were seeded in a 6-well plate. After 24 h of incubation, cells were treated with increasing concentrations of Iscador Qu and Iscador M and incubated for 24 h in complete cell medium. To detach the cells from the culture dish, 0.05% Trypsin-EDTA solution was added to each well. After 5 min of incubation, two volumes of pre-warmed PBS were added to inactivate the trypsinization. The cell suspensions were transferred to tubes and gently centrifuged for 5 min at 900× *g*. After removing the supernatant, the cell pellet was resuspended in pre-warmed PBS. This procedure was repeated 2 times, after which the fluorescent probe was added to the samples to achieve a Laurdan concentration of 5 µM. The dye–cell suspension was incubated for 1 h at 37 °C, 5% CO_2_ in the dark. Then, the cells were centrifuged at 900× *g* for 5 min, the supernatant was removed and prewarmed PBS was added to the pellet. After resuspension of the pellet, the samples were diluted to achieve a cell suspension of 0.5 × 10^6^ cells/mL and transferred to a quartz cuvette to measure the Laurdan emission spectra. The samples were excited at 355 nm and the spectra were recorded in the range of 390 to 600 nm by using an FP-8300 spectrofluorometer (JASCO Ltd., Easton, MD, USA) equipped with a xenon lamp. The temperature in the cuvette holder was maintained at 37 °C by a heating circulating thermostat (Julabo). Data were analyzed by using OriginPro 9.0 software. Laurdan GP values were calculated by using the equation: $$\mathrm{GP}=\frac{\left(I_{440}-I_{490}\right)}{\left(I_{440}+I_{490}\right)}$$
where *I*_440_ and *I*_490_ refer to the emission intensities at those wavelengths. Laurdan GP determines the membrane lipid order. GP values range from −1 to 1, which, respectively, define a membrane in the liquid-disordered phase state (L_d_) when GP tends to −1 and a membrane in the ordered phase state, liquid-ordered (L_o_) or gel phase (L_β_) with GP leaning towards +1.

### 3.6. ξ-Potential Measurements

The ξ-potential of the intact cells and the treated ones was recorded in a cell suspension (0.5 × 10^6^ cells/ml) using the electrophoretic light scattering technique on a Zetasizer Nano ZS analyzer (Malvern Instruments, Malvern, UK). The analyzer uses laser Doppler electrophoresis of cells to measure ξ-potential as a combination of electrophoresis and laser Doppler velocimetry, measuring the speed of cells in PBS upon application of electric field. Once the velocity of the cells is measured the electrophoretic mobility *u* is obtained and ξ-potential is calculated from Helmholtz-Smoluchowski equation [38]. The measurements were performed in a U-shaped cell with gold-plated electrodes at 25 °C and pH 7.4 in PBS. 

### 3.7. Wound Healing (WH) Assay

MCF10A, MCF-7 and MDA-MB231 cells (7 × 10^5^ cells/well) were seeded in a CytoSelect™ 24-well Wound Healing Assay plate containing 12 wound field inserts (Cell Biolabs, Inc., San Diego, CA, USA) and allowed to adhere overnight. After the adherence period, the wound field inserts were removed and the detached cells were washed three times with 500 μL of sterile PBS (1×). Fresh medium (500 μL) containing 75 μmol/mL of tested substances was added afterwards and wound healing was maintained for 24 h. Cell migration was monitored and documented under a phase-contrast microscope Carl Zeiss Jena (Zeiss, Jena, Germany) with 10× magnification at 0 and 24 h post-induction of the wound area. Wound closure was measured and quantified using ImageJ software version 1.8.0_112. The experiment was repeated two times.

### 3.8. Actin Labeling

MCF10A, MCF-7 and MDA-MB231 cells (7 × 10^6^ cells/well) were cultivated on cover glasses (12 × 12 mm) placed in 24-well plates. After 24 h incubation, the cells were treated with Iscador and were incubated for additional 24 h in complete medium. Non-adherent cells were removed by triplicate PBS pH 7.4 rinsing. The adherent cells were fixed with 3% PFA (paraformaldehyde) for 15 min at room temperature. Using 0.5% solution of Triton X-100, the fixed cells were permeabilized and then incubated with 1% BSA (bovine serum albumin) solution. After washing with PBS pH 7.4 three times, BODIPY 558/568 phalloidin was added for 30 min incubation. The samples were installed on objective glasses using Mowiol. Preparations were observed and analyzed using an inverted fluorescent microscope (Leica DM13000, Leica Microsystems GmbH, Wetzlar, Germany) with objective HCX PL FLUOTAR 63x oil.

### 3.9. FACS Analysis

MCF10A, MCF-7 and MDA-MB231 cells (7 × 10^6^ cells/well) were seeded in 14-well plates. After 24 h incubation, they were treated with Iscador and were incubated for another 24 h in full culture media. Non-adherent cells were washed with PBS pH 7.4. Following the incubation, the cells were detached through trypsinization and were labeled with an Anexin V-FITC apoptosis detection kit (Sigma-Aldrich, St. Luis, MO, USA) according to the manufacturer’s instructions. The analysis was performed with BD FACS DIVA software (BD Biosciences, Franklin Lakes, NJ, USA).

### 3.10. Statistical Analysis

All statistical calculations were performed with OriginPro9.0 and GraphPadPrism software (GraphPad Software Inc., SanDiego, CA, USA). Data were analyzed with one-way ANOVA and Tukey–Kramer posttests by t-test where appropriate. Each *p*-value can be considered statistically significant at *p* < 0.05.

## 4. Discussion

In the present study we evaluated the effect of Iscador Qu or M on a set of breast cancer cell lines, as well as their cyto- and photo-toxicity on 3T3/23 cells. The main reason for this test is the possibility that some organic compounds can become phototoxins as a result of light-induced changes in the energy state of the molecule and its degradation. We have shown that the tested substances Iscador Qu and M did not exhibit phototoxicity, making them suitable for subsequent experiments to establish the antiproliferative activity of the mentioned plant extracts. The performed antiproliferative activity test provided information that Iscador Qu and Iscador M possess some selectivity against malignant cells. Both Iscador species exhibit selectivity towards cancer cell lines. The SI values were higher for the MCF-7 cell line compared to MDA-MB-231. The highest SI was found for Iscador Qu (7.7) versus Iscador M (6.9), both for MCF-7 cell lines. The obtained results clearly reveal that Iscador Qu exhibits a stronger antitumor effect compared to Iscador M by reaching 50% antiproliferative effect at concentrations twice as low as compared to Iscador M. This effect was seen in all three cell lines. The observed differences in the action of the two plant extracts are probably due to the differences in the ratios between the active substances of the extracts. Mistletoe contains various bioactive components such as lectin, viscotoxins, and polysaccharides [39,40,41]. The extracts are also enriched in enzymes, sulphurous compounds, fats, flavonoids, phenylpropanes, lignans, alkaloids and others. The most active anticancer compounds are believed to be the lectins and viscotoxins [40,42]. These compounds comprising two groups of toxic proteins as the necrosis is considered to be the primary effect of viscotoxins, whereas lectins are reported to induce apoptotic cell death [19]. Both proteins have been shown to exert immunomodulatory effects. Ulrech et al [43] reported that the total amount of lectins in Iscador Qu was 1.5 times higher than in Iscador M. The authors found that the total amount of lectin was about 261 ± 9.3 ng/mL in 5 mg Iscador M and 391 ± 18.3 ng/mL in 5mg Iscador Qu as the ratio of the three types of lectins ML1/ML2/ML3 is 31/24/45 (%) in Iscador M versus 28/22/50 (%) in Iscador Qu. On the other hand, the total amount of viscotoxins in Iscador Qu is also about 1.5 times higher compared to Iscador M (402.2 ± 17.6 ng/mL for “M” and 607.7 ± 95.4 ng/mL for “Qu”). These data clearly demonstrate that Iscador Qu contains about 1.5 times more active substances, lectins and viscotoxins, than Iscador M, which probably determines its stronger antitumor effect on the investigated cell lines. Moreover, it is assumed that lectins and viscotoxins exert their effect on cancer cells in a symbiotic way. Both, viscotoxins and mistletoe lectins, independently of one another, are cytotoxic compounds, although cell sensitivity depends on the type of the tested cells. For example, Yoshida cells demonstrate high sensitivity to viscotoxin (IC_50_ = 0.7 μg/mL) and less sensitivity to mistletoe lectin (IC_50_ = 12.8 μg/mL). K562 cells, however, are more sensitive to mistletoe lectin (IC_50_ = 75 pg/mL) than Molt4 cells (IC_50_ = 1.3 ng/mL), its high sensitivity to mistletoe lectin being well known [44]. 

Our results reveal that Iscador Qu and Iscador M are able to induce apoptosis for low metastatic MCF-7 cell line unlike control. Furthermore, it was shown that Mistletoe targets two important signaling pathways, P13K/AKT and MAPK, the mechanisms of both pathways being largely discussed in the review of Szurpnicka et al. [27].

Specific differences in the lipid metabolism are reported between cancer and healthy cells. Alterations in the biophysical properties of neoplastic cells may lead to higher resistance to chemotherapy [45]. Cancer cells lipid composition varies from the non-malignant cell profile as there are variations between malignancy types [46,47]. Moreover, the lipid composition of cancer cells may be altered with tumor progression. For example, just before the spread of metastasis, the cancer cell reduces its membrane cholesterol level to increase the membrane fluidity and plasticity needed for penetrating blood vessels [48]. 

In our study, Laurdan stains all cell membranes, plasma membrane as well as all internal membranes. Thus, Laurdan GP values reports the average response of lipid order from all membranes of cell. The lipid order of the membrane is related to mechanical properties of the membrane such as bending and stretching moduli [49]. GP values describe the term fluidity related inversely to the lipid order. The term plasticity of lipid membrane is related to bending and stretching capacity of the lipid membrane with energy dissipation. From biological point of view, the term plasticity reflects larger comprehension. During cancer progression, tumor cells undergo molecular and phenotypic changes collectively referred to as cellular plasticity. Significant cancer cell plasticity, compared to normal cells, is likely associated with the increased epigenomic fluidity in cancer cells [50]. 

We found that the plasma membrane of MCF-10A was more ordered compared to tumorigenic MCF-7 and MDA-MB-231 cells, which is in accordance with the literature reports that the plasma membrane of malignant cells is more fluid compared to non-cancerous cells [51]. The MCF-7 cell line has lower cholesterol level compared to MDA-MB-231 [51,52]. Further, a decreased sphingomyelin (SM) level and an increased level of phosphatidylcholine (PC) species with shorter fatty acyl chain lengths are found in primary cancer cells [53,54] in contrast to non-cancer cells, suggesting more fluid membranes which is in accordance with our results. The highly metastatic cell line MDA-MB-231, however, showed a higher lipid order compared to MCF-7, possibly due to the higher cholesterol content in MDA-MB-231 cells [34]. The presented results herein have shown that MCF10, MCF-7 and MDA MB-231 respond in a specific way, regarding the changes in lipid order induced by Iscador Qu or Iscador M. The direct interaction of Iscador sub-compounds with the plasma membrane of the studied cells probably induces a disorganization of the lipid-protein assemblies which in turn could disturb the cell signaling. Both Iscador species are able to increase the membrane lipid order of control MCF-10A cells with not more than 10 % when treated with IC_50_. For the tumorigenic MCF-7 cell line, the membrane lipid order was decreased by Iscador Qu and increased by Iscador M by about 20% at IC_50_. This implies that Iscador Qu and M interact in a specific way and induce different changes in the membranes of MCF-7 cells. Both, the different lectins/viscotoxins ratio in the studied Iscador species as well as the different SI of both Iscadors to MCF-7 cell line could be the reason for the observed fluidizing or rigidifying effect. Interestingly, both Iscador species did not induce any statistically significant changes in the lipid order of MDA MB-231 membranes in accordance with their higher resistance to Iscador treatment compared to MCF-7 cells.

The interaction of both Iscador species with the analyzed cells was further characterized by ξ-potential measurements. The composition of the plasma membrane and the physiological conditions of cells such as their surrounding environment (molecules and surfaces) generally determine the cell-surface charge. The ξ-potential governs the interaction of the cells with external molecules. This is an electrokinetic potential of the cell at the location of the slipping plane. Both, the untreated and treated with Iscador cells, exhibited negative ζ-potential values as the absolute effect of Iscador Qu treatment was more marked compared to Iscador M. At physiological pH 7.4, the cell ξ-potential for different types of cells is negative and varies within a wide range [55]. This variability could be assigned to differences in the biochemical composition of the plasma membrane. The negative ζ-potential values are supposed to be ascribed to non-ionogenic groups of phospholipids, proteins, and their polysaccaride conjugates [55]. The same authors state that the ξ-potential of dead cells shift towards more negative voltages attributing this absolute ξ-potential increase to the increased level of phosphatidylserine on the cell surface as an early marker of apoptosis. Indeed, the ξ-potential of MCF-7 cells after Iscador Qu treatment showed more negative values compared to Iscador M, which is in accordance with the higher antiproliferative activity observed for the “Qu” compound compared to “M”. Moreover, ζ-potential as well as lipid order changes induced by Iscador treatment are also in accordance with the SI rank established herein: MCF-7 > MDA-MB-231 > MCF-10A. 

The physicochemical changes of the plasma membranes were also accompanied by changes in the cytoskeleton of the tumorigenic lines MCF-7 and MDA-MB-231 after treatment with Iscador Qu or Iscador M. In response to the Iscador treatment the regular organization of the actin filaments was perturbed. Such changes in the organization of the actin cytoskeleton inhibit cell adhesions and affect cell survival. On other hand, the wound healing assay further confirmed the decreased motility and ability of cancer cells to proliferate. The stronger antiproliferative effect of Iscador Qu on the low metastatic cell line MCF-7 compared to Iscador M is possibly due to more significant changes, induced in the membrane lipid order, zeta potential and actin cytoskeleton. 

The presented results demonstrate the potential of white mistletoe plant extract as an antitumor agent, although the exact mechanism of action is still not completely deciphered and needs to be further addressed. Our results clearly demonstrate that *Viscum Album* extracts could have a great potential to sensitize cancer cells to apoptotic cell death if growing concentrations of the chemotherapeutics are applied as it has been recently shown for the first time for MCF-7 cell line [56]. The molecular mechanism of this “trigger” is also of special interest for prevention of a number of ageing-related diseases.

The obtained data as well as these reported in the literature are in accordance and clearly state that both apoptosis/cytotoxic and immunomodulatory activities of mistletoe may contribute to the positive outcome and clinical benefit in breast cancer patients [42,57]. Indeed, clinical trials revealed a beneficial effect on survival, increased health-related quality of life, positive remission rate, and reduction of chemotherapy for breast cancer patients treated with mistletoe extracts [28,58]. In this study, it is shown that both Iscadors are more efficient in low metastatic MCF-7 cell line compared to high metastatic MDA-MB-231 one in *in vitro* experiments. Furthermore, Iscador Q exhibits higher antitumor activity compared to Iscador M. It is noteworthy, there are still no any clinical trials distinguishing the therapeutic potential of both Iscadors on breast cancer cells. 

In conclusion, the present results provide information concerning the biochemical and biophysical mechanisms underlying the influence of Iscador on breast cancer cells in vitro and reveal possibilities for the combined use of *Viscum Album* extracts with specific anti-tumor agents. Further in vivo studies are needed to evaluate the efficacy of Iscador as an adjuvant alternative anti-cancer drug. These studies could serve as a basis for development of complex therapeutic strategies involving mistletoe extracts, which are natural products with well pronounced beneficial properties.

## Figures and Tables

**Figure 1 ijms-24-05247-f001:**
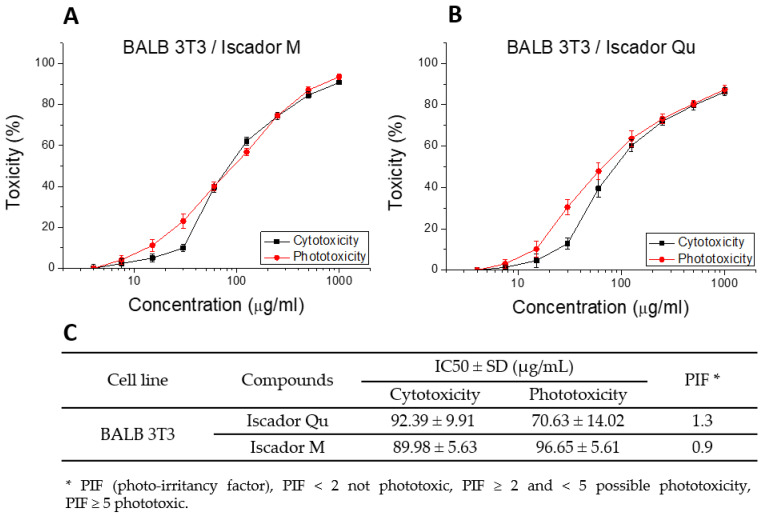
Cytotoxicity and phototoxicity of Iscador M and Qu on BALB 3T3 cells. Results are presented as a percentage viability relative to the control (**A**,**B**). Cell line cytotoxicity and phototoxicity IC50 values summarized in the Table (**C**). Values are means ± SD from three independent experiments.

**Figure 2 ijms-24-05247-f002:**
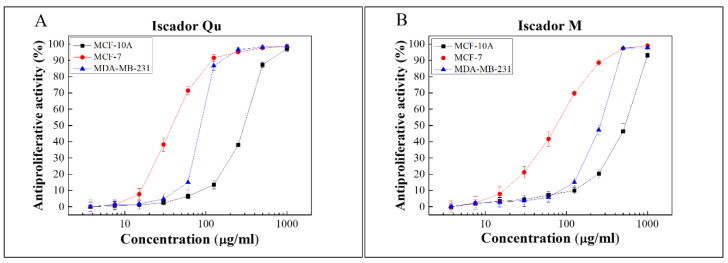
Antiproliferative activity of Iscador Qu (**A**) and Iscador M (**B**).

**Figure 3 ijms-24-05247-f003:**
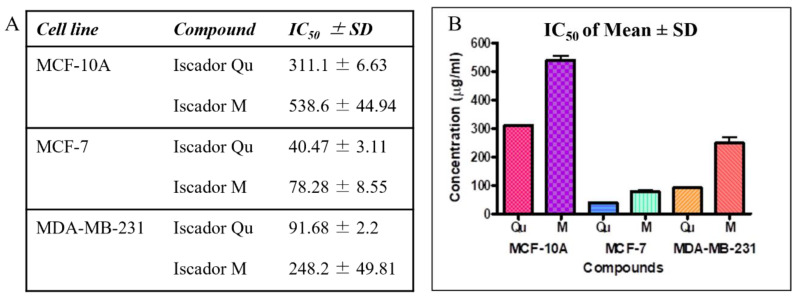
Mean values of the antiproliferative effects (IC_50_) (**A**) and their graphical representation (**B**).

**Figure 4 ijms-24-05247-f004:**
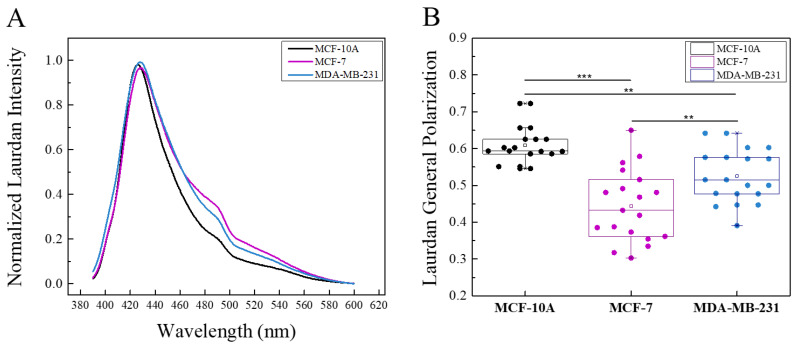
Laurdan fluorescence spectra of untreated tumorigenic cell lines MCF-7 and MDA-MB-231 as well as the non-tumorigenic cell line MCF-10A (**A**). Laurdan GP of untreated non-tumorigenic MCF-10A, tumorigenic MCF-7 and MDA-MB-231 cell lines (**B**). The fluorescence measurements were carried out at 37 °C. The obtained spectrum for one cell line is averaged from total of 18 measurements per cell line. The SD of GP values are obtained from three independent experiments, six measurements per sample. *** *p* < 0.001; ** *p* < 0.01.

**Figure 5 ijms-24-05247-f005:**
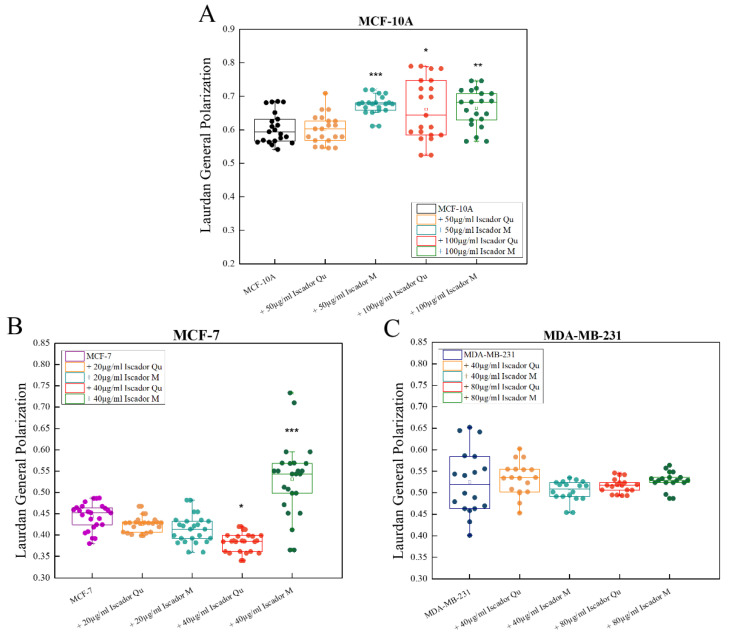
Laurdan GP of untreated and treated with Iscador (Qu and M) tumorigenic MCF-7 and MDA-MB-231 and non-tumorigenic MCF-10A cell lines. All experiments were carried out at 37 °C. (**A**) Laurdan GP of MCF-10A cells without and with Iscador at an IC_50_ concentration of Iscador (100 µg/mL) and a lower one (50 µg/mL); (**B**) Laurdan GP of MCF-7 cells without and with Iscador at an IC50 concentration of Iscador (40 µg/mL) and a lower one (20 µg/mL); (**C**) Laurdan GP of MDA-MB-231 cells without and with Iscador at an IC_50_ concentration of Iscador (80 µg/mL) and a lower one (40 µg/mL). *** *p* < 0.001; ** *p* < 0.01; * *p* < 0.05.

**Figure 6 ijms-24-05247-f006:**
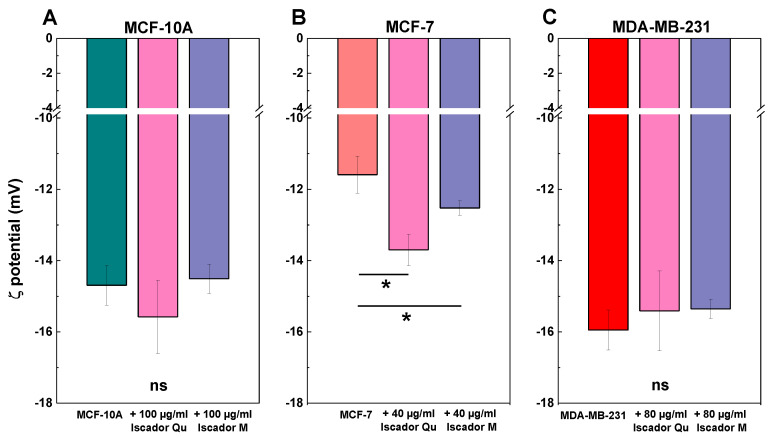
Alterations in ζ-potential of the investigated cell lines, MCF-10A, MCF-7 and MDA-MB-231, treated with Iscador Qu or Iscador M. All ζ-potential values are measured in PBS buffer (137 mM NaCl, 2.7 mM KCl, 8 mM Na_2_HPO_4_ and 2 mM KH_2_PO_4_; pH 7.4). Two independent experiments were carried out for each cell line and each sample is measured six times. *ns* denotes not significant data and * denotes *p* < 0.05.

**Figure 7 ijms-24-05247-f007:**
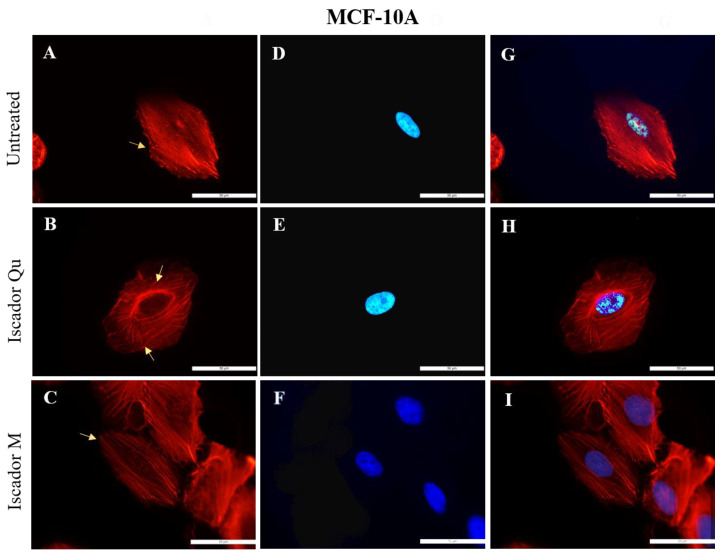
Effect of Iscador on the actin cytoskeleton of MCF-10A cells treated with 100 µg/mL Iscador Qu (IC_50_) or Iscador M and stained with BODIPY/Phaloidin (**A**–**C**); DAPI to label cell nucleus (**D**–**F**); combined cytoskeleton and nucleus (**G**–**I**). Scale bar: 50 µm.

**Figure 8 ijms-24-05247-f008:**
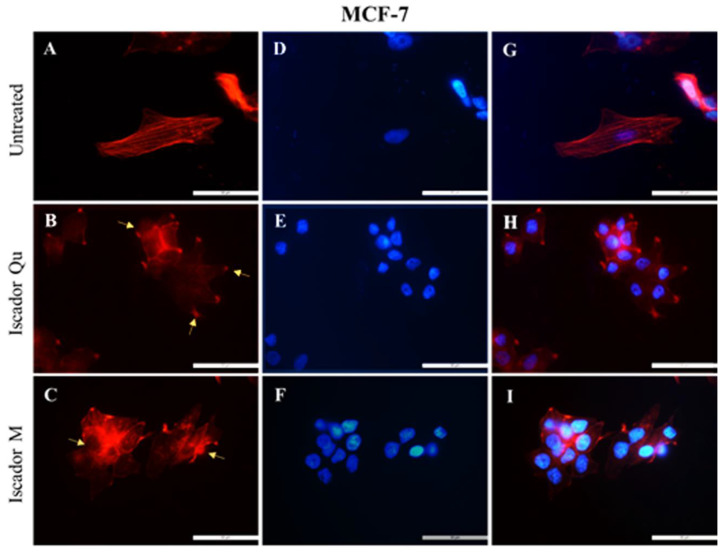
Effect of 40 µg/mL Iscador Qu (IC_50_) or Iscador M on the actin cytoskeleton of MCF-7 cells stained with BODIPY/Phaloidin (**A**–**C**); DAPI (**D**–**F**). Combined cytoskeleton and nucleus (**G**–**I**). Scale bar: 50 µm.

**Figure 9 ijms-24-05247-f009:**
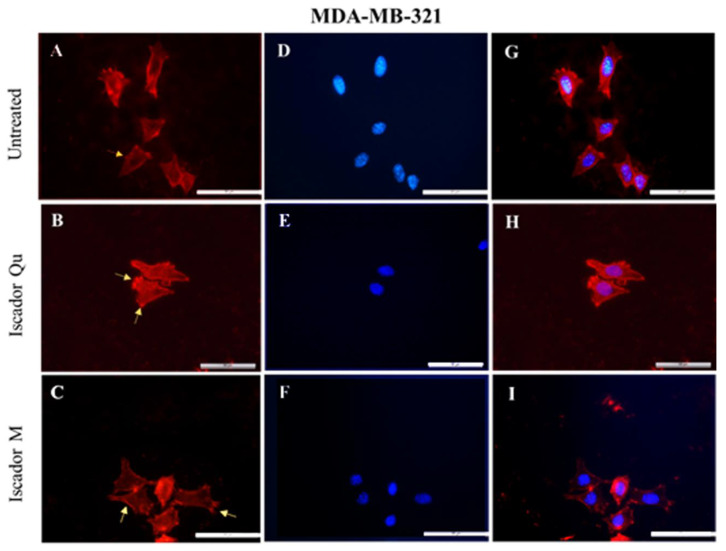
Effect of Iscador on the actin cytoskeleton of MDA-MB-231 cells treated with 80 µg/mL Iscador Qu (IC_50_) or Iscador M and stained with BODIPY/Phaloidin (**A**–**C**); DAPI (**D**–**F**). Combined cytoskeleton and nucleus (**G**–**I**). Scale bar: 50 µm.

**Figure 10 ijms-24-05247-f010:**
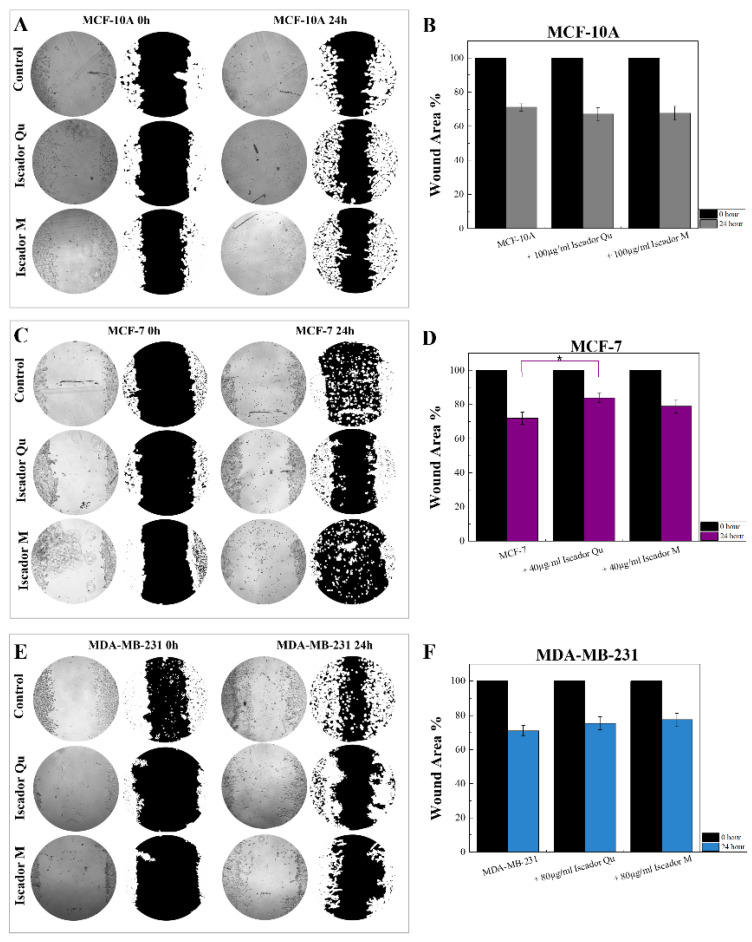
Cell migration assay of non-tumorigenic cells MCF-10A (**A**,**B**), the low metastatic cancer cell line MCF-7 (**C**,**D**) and the high metastatic cancer cell line MDA-MB-231 (**E**,**F**) at 0 and 24 h with and without Iscador Qu or Iscador M. Calculated wound areas at 24 h are presented as a percentage of the initial wound area at 0 h. A statistically significant difference is obtained for Iscador Qu-treated MCF-7 cells compared to control at * *p* < 0.05.

**Figure 11 ijms-24-05247-f011:**
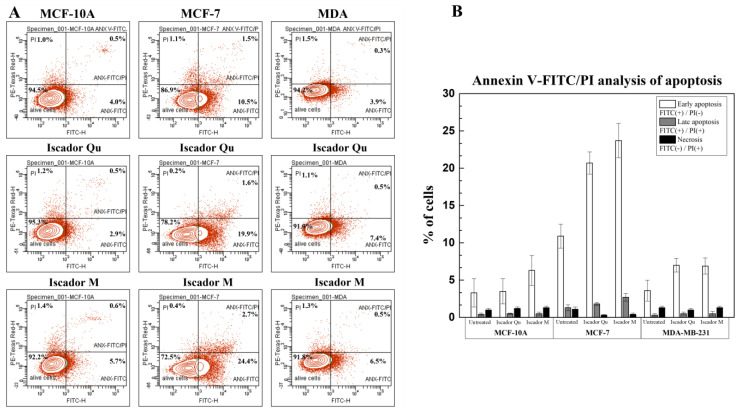
Annexin V-FITC/PI analysis (**A**) of apoptosis in the non-tumorigenic line MCF-10A (left column), and tumorigenic MCF-7 (middle column) and MDA-MB-231 (right column) induced by Iscador Qu or Iscador M; (**B**) FACS analysis data averaged from three independent experiments.

**Table 1 ijms-24-05247-t001:** Selectivity index (SI) of Iscador Qu and Iscador M to malignant cells. SI values calculated against the control cell line MCF-10A.

Selectivity Index (SI)			
MCF-7		MDA-MB-231	
Iscador Qu	Iscador M	Iscador Qu	Iscador M
7.69	6.88	3.39	2.17

## Data Availability

Data is contained within the article.

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
