# Peer review of "Antitumor Effect of Iscador on Breast Cancer Cell Lines with Different Metastatic Potential"

_ijms, 2023, doi:10.3390/ijms24065247_

Round 1

Reviewer 1 Report

There are some minor things that need to be addressed.

1. The MCF10A is a normal cell line so not a low metastatic cell line (Row 91).

2. In row 95 the authors say they tested the extract on three cancer cell lines, but MCF10A cell line is not a breast cancer cell line.

3. In the row 372 something is missing, the author wrote: "(".

4. The authors should also do some supplementary experiments regarding the mechanism of action of these extracts, like apoptosis, analysis and expression analysis of genes or proteins related to apoptosis, cell cycle or EMT mechanism.

Author Response

Reviewer 1:

We would like to thank this reviewer for his/her recommendations, which, no doubt, would contribute to making this paper an interesting source of information.

We have taken into consideration every point raised in this review and have made the required changes as follows:

Review 1

Reviewer 1. The MCF10A is a normal cell line so not a low metastatic cell line (Row 91).

Answer: We agree with the reviewer’s comment that MCF10A is a normal cell line and have corrected the text (line 91). In fact, we have clearly stated that this cell line is non-tumorigenic in line 297, but our statement was not precise especially in line 91.

  1. In row 95 the authors say they tested the extract on three cancer cell lines, but MCF10A cell line is not a breast cancer cell line.

Answer: We have made the required amendments in the text (line 94/95).

  1. In the row 372 something is missing, the author wrote: "(".

Answer: We would like to thank the reviewer for noticing this technical mistake. This was corrected in the revised version of the manuscript.

  1. The authors should also do some supplementary experiments regarding the mechanism of action of these extracts, like apoptosis, analysis and expression analysis of genes or proteins related to apoptosis, cell cycle or EMT mechanism.

Answer: We thank this reviewer for his/her valuable suggestions to analyse in more detail the mechanism of action of the tested extracts. There is ongoing research in our lab, involving the expression of apoptosis-related proteins and some signalling pathways, such as the caspase 3 pathway, which will be subject to another paper, devoted particularly to clarification of the underlying mechanisms.

Reviewer 2 Report

In this study, Robev and colleagues investigate the effect of Iscador Qu (and M) on breast cells. The authors examined the effects of phototoxicity on BALB 3T3 cells. The authors then evaluate the effects of Iscador Qu and M on breast cells and showed those two Iscador have distinct effects growth and IC50(SI) on breast non-tumorigenic or tumor cells. Consistent with these differences, the authors also observed different actin cytoskeleton phenotype and Wound healing assay upon treatment with Iscador Qu and M. Lastly, the authors showed that apoptosis of treatment Iscador Qu and M using Annexin V/PI staining. Overall, the author needs to clarify the class of cancer cell death and apoptosis induced by this study to gain insight into the anticancer activity of Iscador. There are several issues need to be addressed to better support fundamental information on cancer cell death induced by Iscador.

Q1: Iscador Qu was investigated by phototoxicity test in Figure1. Did the authors try any of Iscador M in NRU 3T3 test?

Q2: (In Figure 5, 6.) The change of lipid order and zeta potential in only MCF-7 cells upon treatment with Iscador Qu and M are shown. Please clarify fluidity and plasticity are involved in Gp values in MCF-10A and MDA-MB-231 cells.

Q3: In Figure 7 - 9, change of actin cytoskeleton and wound healing assay in three cell lines upon treatment with Iscador are shown. It is also desirable to clarify the change of cellular distribution of EMT markers such as N-cadherin, Vimentin, and Zeb… etc.

Q4. Iscador Qu have been reported to affect ERK and p38 signaling in MCF-7 breast cancer cell. The author should test the effect of mechanism of action in MCF-7 and MDA-MB-231 cells, at least in the gene or protein level.

Q5. In Figure 11, induced apoptosis of cells with Iscador did not in correlation with from the viability test. Also, Iscador M is showed stronger apoptosis activity compared to Iscador Qu in MCF-7 cells. The authors should distinguish the effects on those two aspects in their assays.

Q6. The author needs to clarify the class of apoptosis induced by this Iscador to gain insight into the molecular mechanism (such as Bax, Bcl2, caspase-3…) of anticancer activity.  

Reviewer 3 Report

The research article presents the effects of two commercially available Viscum album extracts, Iscador Qu and Iscador M, on three breast cancer cell lines. Iscador Qu demonstrated higher anti-proliferative and anti-migratory activity toward the low metastatic cell line, MCF-7, in comparison to Iscador M. An increase in apoptotic cell population was observed upon treatment with the extract, and changes in the zeta potential and membrane lipid order were observed. Overall, the subject of the research is interesting, however, some discrepancies between the results and conclusions need to be addressed. Below are my comments for the authors:

-In the cell migration assay the authors observed the anti-migratory activity of the extracts at their IC50 concentration. Thus, the cytotoxicity of the extracts could influence the observed results. I recommend determining the anti-migratory potential of the extracts at lower concentrations with extended time periods.

- Without statistical analysis of the cell migration assay, conclusions cannot be drawn. The authors mention that statistical analysis was performed, however, no standard deviations or statistical significance is indicated on the graphs. Furthermore, in Figure 10, from the panel with images of wound closure, it seems as though the wound area percentage is higher upon treatment with Iscador Qu in MDA-MB-231 cells in comparison to control cells (24 h) than in MCF-7 cells, which is not in line with the graphs. Moreover, controls in the MDA-MB-231 panel (0 h) are not uniform and could influence the results. Thus, these results should be revised.

- What was the solvent used for the Iscador Qu and Iscador M extracts, and were control cells treated with the same solvent?

- The number of repetitions of the FACS analysis used to determine apoptosis induction should be provided along with the statistical analysis.

- It would be interesting to the reader to provide information regarding previously performed in vivo analysis of these commercially available preparations.

Round 2

Reviewer 2 Report

I believe the authors have addressed all my previous concerns. The revised manuscript has been improved based on the changes. 

Author Response

We have taken into consideration all reviewer's remarks.

We have replaced the table in Fig.11 with a bar graph according to reviewer 3 suggestion.

Best regards

Reviewer 3 Report

The authors have addressed my comments. However, I suggest replacing the added table in Figure 11 with a bar graph. Such a representation of the results would be more clear.

Author Response

We replaced the added table in Figure 11 with a bar graph according to the reviewer's remark. We agree that the new representation of the results is more clear.

We thank the reviewer for this suggestion.

Best regards